# Distinct regional meteorological influences on low cloud albedo susceptibility over global marine stratocumulus regions

Jianhao Zhang[1,2] and Graham Feingold[2]

[1]Cooperative Institute for Research in Environmental Sciences (CIRES), University of Colorado, Boulder, CO, USA
[2]Chemical Sciences Laboratory, National Oceanic and Atmospheric Administration (NOAA), Boulder, CO, USA

**Correspondence:** Jianhao Zhang (jianhao.zhang@noaa.gov)

**Abstract.** Marine stratocumuli cool the Earth effectively due to their high reflectance of incoming solar radiation and persistent occurrence. The susceptibility of cloud albedo to droplet number concentration perturbations depends strongly on large-scale meteorological conditions. Studies focused on the meteorological dependence of cloud adjustments often overlook the covariability among meteorological factors and their geographical and temporal variability. We use 8 years of satellite observations sorted by day and geographical location to show the global distribution of marine low cloud albedo susceptibility. We find an overall cloud brightening potential for most marine warm clouds, more pronounced over subtropical coastal regions. A weak cloud darkening potential in the annual mean is evident over the remote SE Pacific and SE Atlantic. We show that large-scale meteorological fields from the ERA5 reanalysis data, including lower-tropospheric stability, free-tropospheric relative humidity, sea surface temperature, and boundary layer depth, have distinct covariabilities over each of the eastern subtropical ocean basins where marine stratocumuli prevail. This leads to a markedly different annual cycle in albedo susceptibility over each basin. Moreover, we find that basin-specific regional relationships between key meteorological factors and albedo susceptibilities are absent in a global analysis. Our results stress the importance of considering the geographical distinctiveness of temporal meteorological covariability when scaling up the local-to-global response of cloud albedo to aerosol perturbations.

## 1 Introduction

Marine warm (liquid) clouds cover about one third of the global ocean surface in annual mean (Chen et al., 2014). They prevail over low-latitude to mid-latitude oceans, more pronouncedly over the eastern subtropical oceans where the Earth's major semi-permanent marine stratocumulus decks form (Klein and Hartmann, 1993; Wood, 2012). These bright and blanket-like stratiform clouds reflect a good fraction of the incident solar radiation (ranging from 0.35 to 0.42 in annual mean; Bender et al., 2011) that would otherwise (in the absence of these clouds) be largely absorbed by the dark ocean ($\sim$94%), effectively cooling the Earth (e.g. Stephens et al., 2012). For warm clouds exhibiting constant macrophysical properties (e.g., liquid water path (LWP) and cloud cover), their brightness, or cloud albedo ($A_c$), quantified as the ratio of the reflected shortwave flux to the incoming solar radiation at the top of atmosphere, is particularly sensitive to the droplet concentration ($N_d$), such that higher $N_d$ accompanied by smaller drops makes the cloud more reflective (cloud brightening; Twomey, 1974, 1977). However, cloud macrophysical properties do change with time as the cloud system evolves, through precipitation, evaporation, and/or

entrainment mixing processes (Wood, 2012). Microphysical changes in $N_d$ and droplet sizes induced by aerosol perturbations can substantially modulate the rate and efficiency of these processes and thereby cause further adjustments in macrophysical properties and cloud albedo (e.g. Boucher et al., 2013; Ackerman et al., 2004; Bretherton et al., 2007; Jiang et al., 2006).

In nature, the responses of cloud macrophysical properties to $N_d$ perturbations are always complicated by the variability driven by local meteorology, and for decades, the stated challenge and focus has been to untangle aerosol effects from covary-
ing meteorological conditions (Stevens and Feingold, 2009). Simulations of marine boundary layer (MBL) clouds, in which meteorology can be easily controlled, indicate a bidirectional LWP adjustment to increasing $N_d$. For precipitating clouds, an increase in $N_d$ induces smaller droplets that suppress condensate removal, eventually leading to an increase in LWP (brighter clouds; Albrecht, 1989). In the case of non-precipitating clouds, the reduced droplet sizes lead to weaker sedimentation fluxes at cloud tops (Bretherton et al., 2007) and faster evaporation (Wang et al., 2003; Xue and Feingold, 2006), which both cause
stronger entrainment mixing that reduces cloud LWP, resulting in less reflective clouds.

Observations of cloud adjustments following anthropogenic aerosol perturbations confirm the bidirectional LWP responses (e.g. Chen et al., 2012; Trofimov et al., 2020), while the aggregated response remains uncertain (Malavelle et al., 2017; Toll et al., 2019; Christensen et al., 2022). This means that cloud LWP responses to increased $N_d$ can either enhance or offset the microphysical brightening depending on the meteorological conditions. Progress has been made over the years towards estab-
lishing fundamental knowledge of the environmental state/regime dependence of cloud adjustments to aerosol perturbations. For inversion-capped MBL clouds, the budget of cloud condensate is regulated mainly by entrainment drying at cloud tops and the fraction of precipitation that reaches the surface, which are strongly dependent on the humidity in the free troposphere and the lower-tropospheric stability (Ackerman et al., 2004; Chen et al., 2014; Gryspeerdt et al., 2019). In part related to the atmospheric stability, clouds exhibit a much more negative LWP response to increased $N_d$ in deep MBLs than those that reside
in shallower MBLs (e.g. Possner et al., 2020; Toll et al., 2019). Furthermore, Dagan et al. (2015) show that the direction in which cloud condensate responds to an increase in aerosol depends on an optimal aerosol concentration which is determined by thermodynamic conditions (temperature and humidity). Wood (2007) shows that cloud-base height is the key determinant of whether cloud thickness changes will enhance or offset the Twomey brightening.

Clearly, the spatiotemporally scaling up (e.g. local-to-global and/or transient-to-climatology) of cloud albedo responses to
aerosol perturbations depends crucially on the frequency of occurrence of the environmental states that characterize cloud adjustments. However, the spatiotemporal distribution of the covariability between meteorological and aerosol conditions is understudied and often ignored in "untangling" studies. Mülmenstädt and Feingold (2018) state the need for a shift in attention from untangling aerosol effects from covarying meteorology towards embracing and understanding the covariabilities between them. The focus of this study is exactly on this point.
Using 8 years of satellite observations and the ERA5 reanalysis dataset (introduced in Section 2), we characterize the geographical distribution of marine warm cloud albedo susceptibility over global oceans from 60° S to 60° N (Section 3). We show that similar free-tropospheric and boundary layer conditions lead to different albedo susceptibilities in different stratocumulus basins (Section 4), attributed to the distinct temporal covariabilities among large-scale meteorological conditions (Section 5.1). We find distinct monthly evolutions of albedo susceptibility in different stratocumulus basins, covarying with

each basin's low cloud frequency of occurrence and aerosol conditions (Section 5.2). We conclude that a frequency-weighted global aggregation of albedo-susceptibility-meteorology relationships obscures regionally distinct features and thus provides a biased view on the environmental dependence of albedo susceptibility.

## 2    Data and Methods

    We obtain coincident marine low-cloud properties, including cloud optical depth ($\tau$), cloud top effective radius ($r_e$), low-cloud
fraction ($f_c$), cloud LWP, cloud top height (CTH), and top-of-atmosphere (TOA) shortwave (SW) fluxes from the MODerate resolution Imaging Spectroradiometer (MODIS) (Platnick et al., 2003) and the Clouds and the Earth's Radiant Energy Systems (CERES; Wielicki et al., 1996) sensors onboard the Terra and Aqua satellite (overpass $\sim$10:30 and $\sim$13:30 local time, respectively), which are integrated into the CERES Single Scanner Footprint (SSF) product Edition 4 (level 2) with a footprint resolution of 20 km (Su et al., 2015). $N_d$ is calculated following Zhang et al. (2022) for all CERES footprints with cloud
effective temperature greater than 273 K, CTH less than 3 km, $\tau > 3$, $r_e > 3$ $\mu$m, solar zenith angle $< 65°$, and $f_c > 0.8$, in order to minimize retrieval biases (Grosvenor et al., 2018; Painemal et al., 2013; Grosvenor and Wood, 2014). Footprint cloud properties are aggregated to $1°$ spatial resolution, using only above mentioned cloudy footprints where $N_d$ is retrieved, to match susceptibilities calculated for individual $1° \times 1°$ satellite snapshots. At this scale, the confounding effect of meteorology on footprint (i.e. sub-$1°$) cloud properties is negligible (Goren and Rosenfeld, 2012, 2014). Therefore, linear least-squares log-log
regressions of footprint properties are used to calculate albedo susceptibility $S_0 = d\ln(A_c)/d\ln(N_d)$ and radiative susceptibility $F_0 = d(A_c)/d\ln(N_d) \times f_c \times \mathrm{SW_{dn}}$; for both metrics, positive values indicate more reflected sunlight, thereby cooling, following Zhang et al. (2022). Note that since we are interested in the overall response of $A_c$ to $N_d$ perturbations, including both micro- and macro- physical adjustments, we do not stratify by LWP when calculating $S_0$. The result is that we obtain susceptibilities that account for both the Twomey effect and adjustments of cloud LWP. This differentiates our method from
that of Painemal (2018) and Rosenfeld et al. (2019). Moreover, calculating $S_0$ from individual satellite snapshots enables us to associate the derived $S_0$ to the mean meteorological and cloud states of individual cloud scenes at a given space and time. The logarithmic transformation alleviates the dependence of $S_0$ on the absolute value of $N_d$, minimizing the impact of the remaining $N_d$ retrieval biases (e.g. due to the adiabatic assumption). We do, however, use absolute values of $A_c$ in the $F_0$ expression to obtain flux responses in units of W m$^{-2}$, instead of percentage responses.
Meteorological conditions, including sea surface temperature (SST), lower tropospheric temperature, humidity, and wind profiles, are obtained from the European Centre for Medium-Range Weather Forecasts (ECMWF) fifth-generation atmospheric reanalysis (ERA5; Hersbach et al., 2020), and interpolated and aggregated to the Terra and Aqua overpass times at $1°$ spatial resolution. Lower-tropospheric-stability (LTS) is calculated as the difference in potential temperature between 700 hPa and 1000 hPa. Free-tropospheric relative humidity (RH$_{ft}$) is defined as the the mean relative humidity between inversion top (defined
as the level of the strongest gradient in temperature and humidity) and 700 hPa, following Eastman and Wood (2018).

    The datasets span $60°$ S to $60°$ N, covering global oceans, from 2005 to 2012 (8 years). We screen for cloudy satellite scenes over open water when only single layer liquid cloud (SLLC) is present. Aqua and Terra observations are analyzed separately,

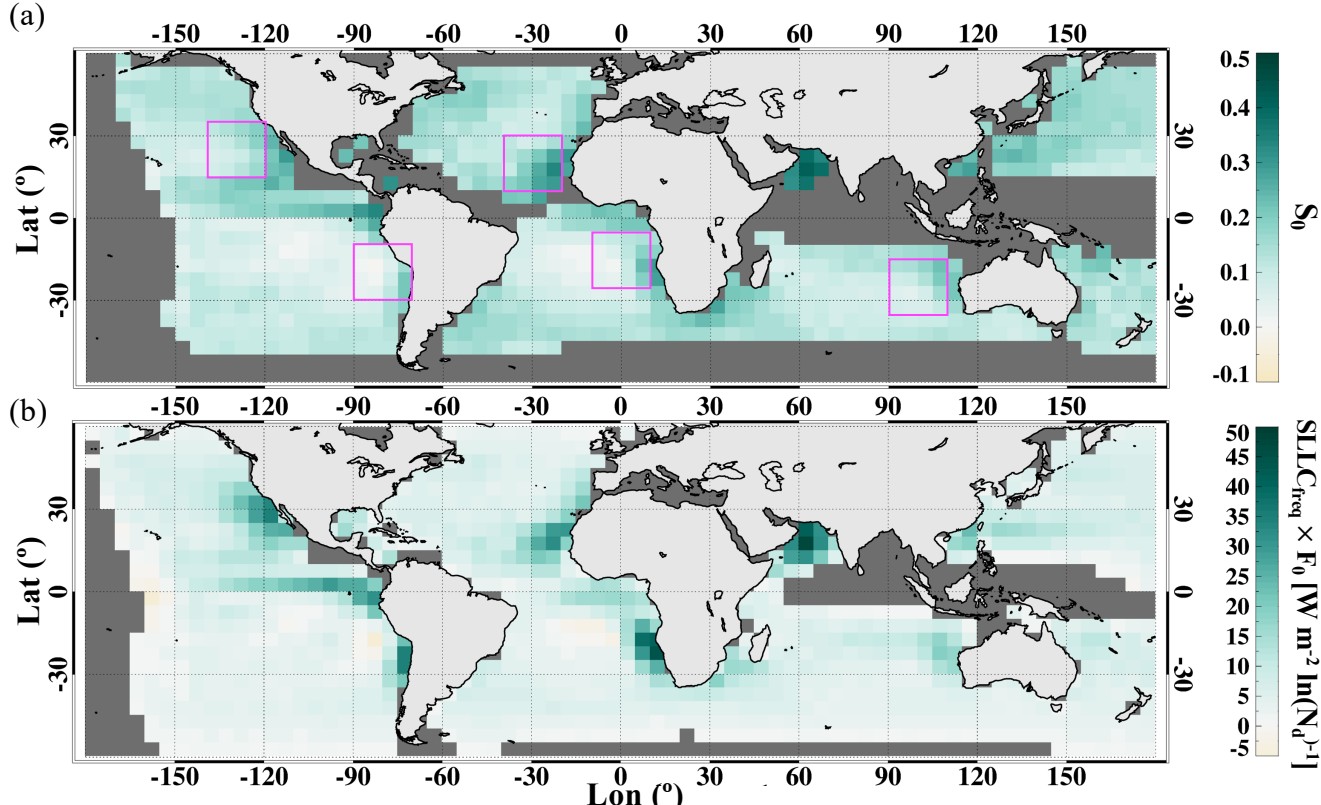

**Figure 1.** Geographical distribution of marine single layer low-cloud (a) albedo susceptibility ($S_0$) and (b) radiative susceptibility ($F_0$) weighted by the frequency of occurrence of single layer liquid cloud (SLLC). Spatial-temporal averages of $5° \times 5°$ areas are shown. Only areas with SLLC frequency of occurrence greater than 0.1 are shown in (a). Magenta boxes in (a) indicate five $20° \times 20°$ marine stratocumulus regions analyzed further in this study

instead of collectively, in order to assess robustness of our findings (qualitatively) and to explore the role of the diurnal cycle (quantitatively). Analyses using the Aqua observations are shown in the main text, and those using the Terra observations are shown in the supplementary material, given the similarity between Aqua and Terra results. Regional annual maxima in SLLC fractional coverage and frequency of occurrence are used to identify 5 major marine stratus/stratocumulus regions ($20° \times 20°$, Fig. S1, magenta boxes).


## 3  Global distribution of marine low cloud albedo susceptibility

### 3.1  Annual mean

The climatology of geographical distribution of marine low-cloud $S_0$ (Fig. 1a) is represented by an aggregation of susceptibilities derived from individual satellite snapshots over the 8-year period, taking into account the frequency of occurrence of different cloudy scenes and meteorological regimes. It is clear that over most parts of the global ocean ($60°$ S to $60°$ N), low clouds have a positive $S_0$ (brightening) in the annual mean, more pronouncedly off the coast of continental land masses although $N_d$ is climatologically higher (Fig. S2) and the MBL is shallower (Fig. S3), compared to those over remote oceans.

Only over the remote subtropical southeast Pacific/Atlantic regions, do the data show weak darkening potential (negative $S_0$) in the annual mean. The darkening potential means that the brightening of the clouds via the Twomey effect – i.e. more particles lead to more droplets and brighter clouds – is more than compensated by liquid water losses.

One can then translate the $S_0$ map into an annual flux perturbation potential map (Fig. 1b), which highlights the high annual cooling potential over subtropical stratocumulus regions even more, by taking into account the cloud fraction and frequency and amount of incoming solar radiation at a given geographical location. In the remote parts of the subtropical stratocumulus

decks, warmer SSTs deepen the MBL and encourage entrainment of subsiding free-tropospheric air at cloud tops (Fig. S3 and Bretherton, 1992; Wyant et al., 1997), favoring entrainment-feedback-driven LWP decreases with increasing $N_d$ (Bretherton et al., 2007; Wang et al., 2003). The location where this MBL condition prevails is consistent with the location where we observe cloud darkening potential offsetting the Twomey brightening potential in the annual mean, resulting in a net warming

potential over the southeast Pacific/Atlantic (Fig. 1b).

### 3.2  Brightening versus darkening regimes

The LWP–$N_d$ variable space has been shown as a useful framework to infer process-level understanding of aerosol-cloud interactions, using satellite observations (e.g. Zhang et al., 2022) or cloud-resolving simulation outputs (e.g. Glassmeier et al., 2019; Hoffmann et al., 2020). Here we show $S_0$ in the LWP–$N_d$ variable space for the five major stratocumulus regions (Fig.

2), similar to Fig. 3 in Zhang et al. (2022), where 3 clearly separated susceptibility regimes are evident. Note although $S_0$ is shown for each LWP–$N_d$ bin, the calculation of $S_0$ applied to individual Terra/Aqua snapshots is not stratified by LWP (see details in Section 2), and thereby, both LWP adjustments (shown in Fig. S4) and the Twomey effect contribute to these albedo susceptibilities at different LWP–$N_d$ states. The 3 regimes are:

1. *Precipitating brightening.* This regime consists of clouds with larger droplets (i.e. high cloud liquid water but low droplet

number, i.e. to the left of the 12 $\mu$m isolines) that are likely to precipitate (Gerber, 1996; vanZanten et al., 2005). A positive susceptibility (brighter clouds with higher $N_d$) is consistent with the Twomey effect (microphysical adjustment; Twomey, 1974, 1977) and the precipitation-suppression induced lifetime effect (macrophysical adjustment; Albrecht, 1989). Both effects contribute to the cloud brightening potential of this regime, and we do not attempt to separate them,

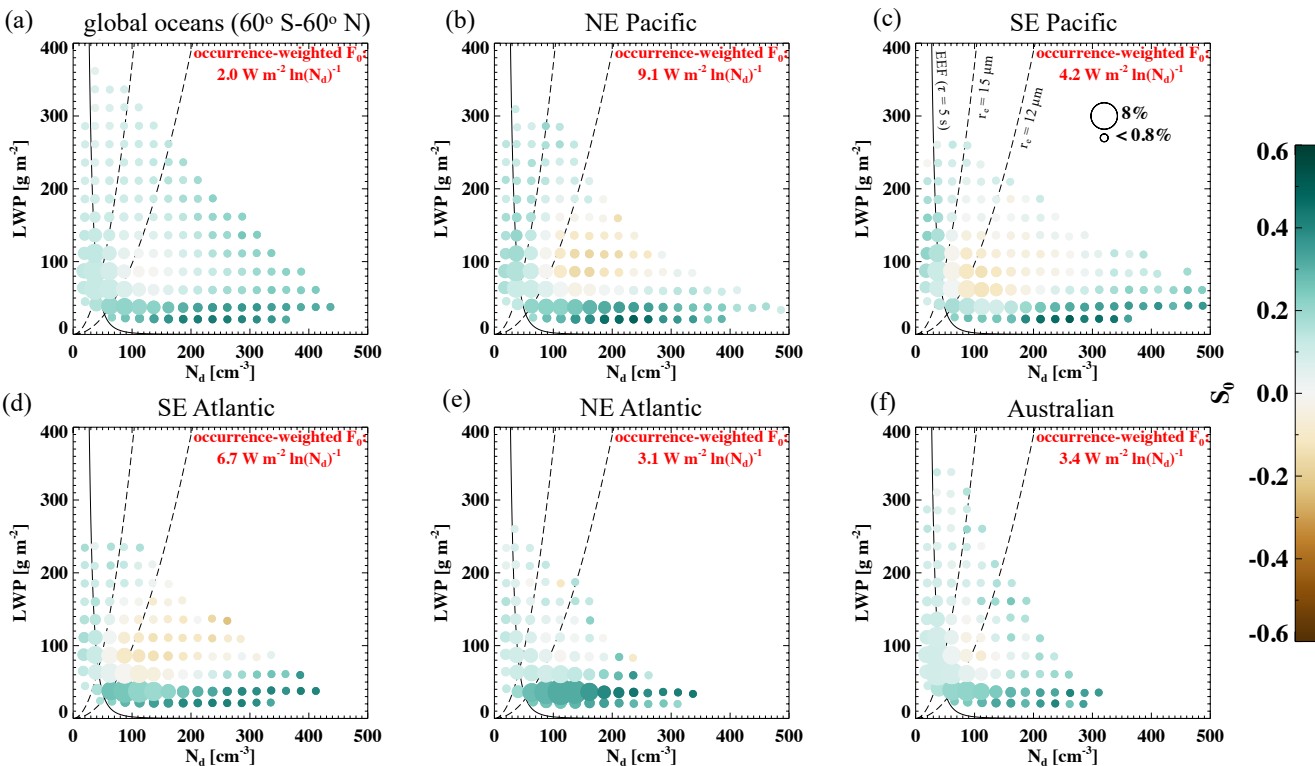

**Figure 2.** Cloud albedo susceptibility ($S_0$, colored filled circles) in LWP–$N_d$ variable space, as bin means (bin size of 25 g m$^{-2}$ and 25 cm$^{-3}$), for (a) global oceans (60° S - 60° N), and 20° × 20° boxes over (b) NE Pacific, (c) SE Pacific, (d) SE Atlantic, (e) NE Atlantic, and (f) Australian stratocumulus regions. The size of the circles indicates the frequency of occurrence of a LWP-$N_d$ bin (reference circle sizes with corresponding occurrence are indicated on panel (c)). Bins with less than 0.01% frequency of occurrence (or less than 25 samples) are not shown. Isolines of evaporation–entrainment feedback (EEF; phase relaxation timescale of 5 s) and isolines of $r_e$ of 12 and 15 $\mu$m based on an adiabatic condensation rate of $2.14 \times 10^6$ kg m$^{-4}$ (black dashed; commonly used measures of precipitation) are indicated on panel (c). Mean radiative susceptibility ($F_0$) weighted by the frequency of occurrence of each LWP–$N_d$ bin and the effective cloud frequency (freq$_{\mathrm{eff}}$, defined in Eqn. 1) is printed in red (named occurrence-weighted $F_0$).

as "untangling" is not the goal of this study. The focus on high-$f_c$ scenes removes scenes with heavy precipitation, which is desirable since precipitation represents a cloud/rain effect on aerosol rather than an aerosol effect on cloud albedo.

2. *Darkening.* For clouds that are not heavily precipitating, cloud top entrainment drives a negative tendency in cloud LWP that can overcome the positive tendencies driven by LW cooling and surface fluxes and lead to cloud thinning and/or breakup (Hoffmann et al., 2020). A decrease in $r_e$ and increase in $N_d$ (assuming constant LWP when clouds respond microphysically to aerosol perturbations) lead to an increase in the overall droplet surface area, enhancing droplet evaporation; meanwhile, sedimentation fluxes at cloud tops reduce with smaller droplets. Enhanced evaporation and reduced sedimentation at cloud tops cause stronger entrainment mixing which further enhances evaporation and reduces sedimentation, creating positive feedback loops, termed as the entrainment-evaporation feedback (EEF; Wang et al., 2003; Xue and Feingold, 2006) and the sedimentation-entrainment feedback (SEF; Bretherton et al., 2007), respectively. Moreover, SW heating during daytime, which can also be enhanced by increasing $N_d$ and decreasing $r_e$, contributes to cloud thinning and/or breakup as well (Petters et al., 2012). Cloud states showing both negative LWP adjustment (Fig. S4) and negative $S_0$ (darker clouds with higher $N_d$) are evident in 4 of the 5 stratocumulus regions for thicker clouds (LWP > 50 g m$^{-2}$) with higher droplet number concentration ($N_d > 50$ cm$^{-3}$) (Fig. 2b-d, f). This indicates that the Twomey effect is more than compensated by these cloud thinning processes that can be enhanced by a reduction in droplet sizes. Here we exclude the possibility of aerosol direct and semi-direct effects driving the cloud darkening, as the $S_0$ shown in this study is calculated at $1°$ resolution, a scale at which we expect overlying absorbing aerosols to be spatially homogeneous.

3. *Non-precipitating brightening.* For thinner non-precipitating clouds (LWP < 50 g m$^{-2}$), cloud-top entrainment efficiency is much reduced, compared to thicker clouds (Hoffmann et al., 2020), and LW cooling surpasses SW heating at cloud tops (Petters et al., 2012), making it easy for the Twomey effect to overcome the cloud thinning processes and dominate the $A_c$ response. Moreover, weak positive correlations between LWP and $N_d$ are observed for these non-precipitating thin clouds (Fig. S4). Although this contributes to the overall brightening potential, it does not necessarily represent a LWP adjustment to $N_d$ changes. In fact, LWP may be the driver of this relationship when entrainment processes take a minor role in regulating LWP. In such a case, clouds with higher LWP generate stronger cooling and turbulence at cloud tops, which helps activate more cloud droplets. As a result, values of $S_0$ and $F_0$ in this regime might be slightly overestimated, making these satellite estimates an upper-bound of the true susceptibilities.

Although different stratocumulus basins have different cloud state distributions in the LWP–$N_d$ variable space, the non-precipitating brightening and the precipitating brightening regimes remain rather persistent – that is cloud states to the left of the 12-15 $\mu$m isolines and cloud states with LWP < 50 g m$^{-2}$, respectively. In contrast, cloud states associated with a darkening potential vary from basin to basin, from almost absent over the NE Atlantic (occurring ~2% of the time) to occurring ~32% of the time over the SE Atlantic and ~34% over the SE Pacific (Fig. 2). This sensitivity of the darkening regime to ocean basin (discussed further in the following sections) is consistent with a dependence of LWP adjustment on meteorological conditions (e.g. Zhang et al., 2022; Gryspeerdt et al., 2019; Possner et al., 2020). When all marine low clouds are combined in the LWP–

$N_d$ space (Fig. 2a), one may conclude a predominant cloud brightening potential and a lack of cloud darkening potential, which is not the case for 3 of the Earth's major semi-permanent stratocumulus decks (Fig. 2b-d).

To learn how these susceptibility regimes constitute the overall albedo susceptibility of each geographical location, we quantify the frequency of occurrence (Fig. 3) of each susceptibility regime and their contributions to the overall $F_0$ (Fig. 4) for $5° \times 5°$ oceanic areas globally, based on the sign of $S_0$ and an $r_e$ of 12 $\mu$m (above which clouds are more likely to drizzle) as manifested in the LWP–$N_d$ variable space. Clearly, the three susceptibility regimes have distinct geographical preferences (Fig. 3 and 4). The non-precipitating brightening regime occurs the most frequently over the shallow, often polluted, stratus/stratocumulus off the coast of continents and tends to dominate the $F_0$ therein (Fig. 3a and 4a). The precipitating brightening regime, although occurring over 50% of the time over most parts of the remote, clean oceans and the equatorial eastern Pacific (Fig. 3c), contributes little to the overall $F_0$ (Fig. 4c), due to the low areal coverage of these often precipitating clouds (disorganized or open-cellular form). In between the geographical preferences of the above two regimes lies the region where the darkening regime (mostly non-raining) becomes the leading contributor to the overall $F_0$, especially over the SE Pacific and Atlantic (Fig. 3b and 4b), where net warming potentials are observed (Fig. 1).

### 3.3 Scale-up of the $F_0$ assessment

The radiative susceptibility derived in this study, $F_0$, which represents a TOA SW flux response to a unit change in $\ln(N_d)$, is equivalent to the radiative sensitivities shown in Bellouin et al. (2020). In order to compare the values from both studies, we scale the aggregated (in LWP–$N_d$ space) $F_0$ (derived from high-$f_c$ liquid cloud scenes) by an effective cloud frequency (freq$_{eff}$) term that takes into account the spatial covariability between $F_0$ and the frequency of occurrence of high-$f_c$ scenes (scene$_{freq}$), which takes the form:

$$\text{freq}_{eff} = \frac{\langle \text{scene}_{freq} \times F_0 \rangle}{\langle F_0 \rangle}, \tag{1}$$

where angle brackets denote spatial averaging. This freq$_{eff}$ scaled $F_0$, which we call the occurrence-weighted $F_0$, is denoted on Fig. 2 and Fig. S6 for global marine low clouds and individual stratocumulus/stratus basins, using Aqua and Terra observations, respectively. For global marine low clouds, we obtain an occurrence-weighted $F_0$ of 2.0 W m$^{-2}$ ln($N_d$)$^{-1}$ (freq$_{eff}$ being 0.061) using Aqua observations, and 3.1 W m$^{-2}$ ln($N_d$)$^{-1}$ (freq$_{eff}$ being 0.082) using Terra observations. These are within the bounds provided in Bellouin et al. (2020) based on multiple lines of evidence, when one sums up their radiative sensitivities for the Twomey effect ($S_{\mathcal{N}} c_{\mathcal{N}}$) and the rapid adjustment in LWP ($\beta_{\ln(\mathcal{L})-\ln(\mathcal{N})} S_{\mathcal{L},\mathcal{N}} c_{\mathcal{L}}$) (using their notations). Bellouin et al. (2020) obtain a cooling sensitivity of $\sim$3.8 W m$^{-2}$ ln($N_d$)$^{-1}$ when central values from their Table 4 are used. The difference with the results shown here likely stems from the focus of our study on marine-only high-$f_c$ scenes, whereas Bellouin et al. (2020) integrate evidence from both modeling and satellite studies that cover global warm clouds. When comparing to Bellouin et al. (2020), we do not include the rapid adjustment in cloud cover term, as our study does not consider cloud fraction changes due to $N_d$ perturbations. Also, note the difference in sign convention between the two studies.

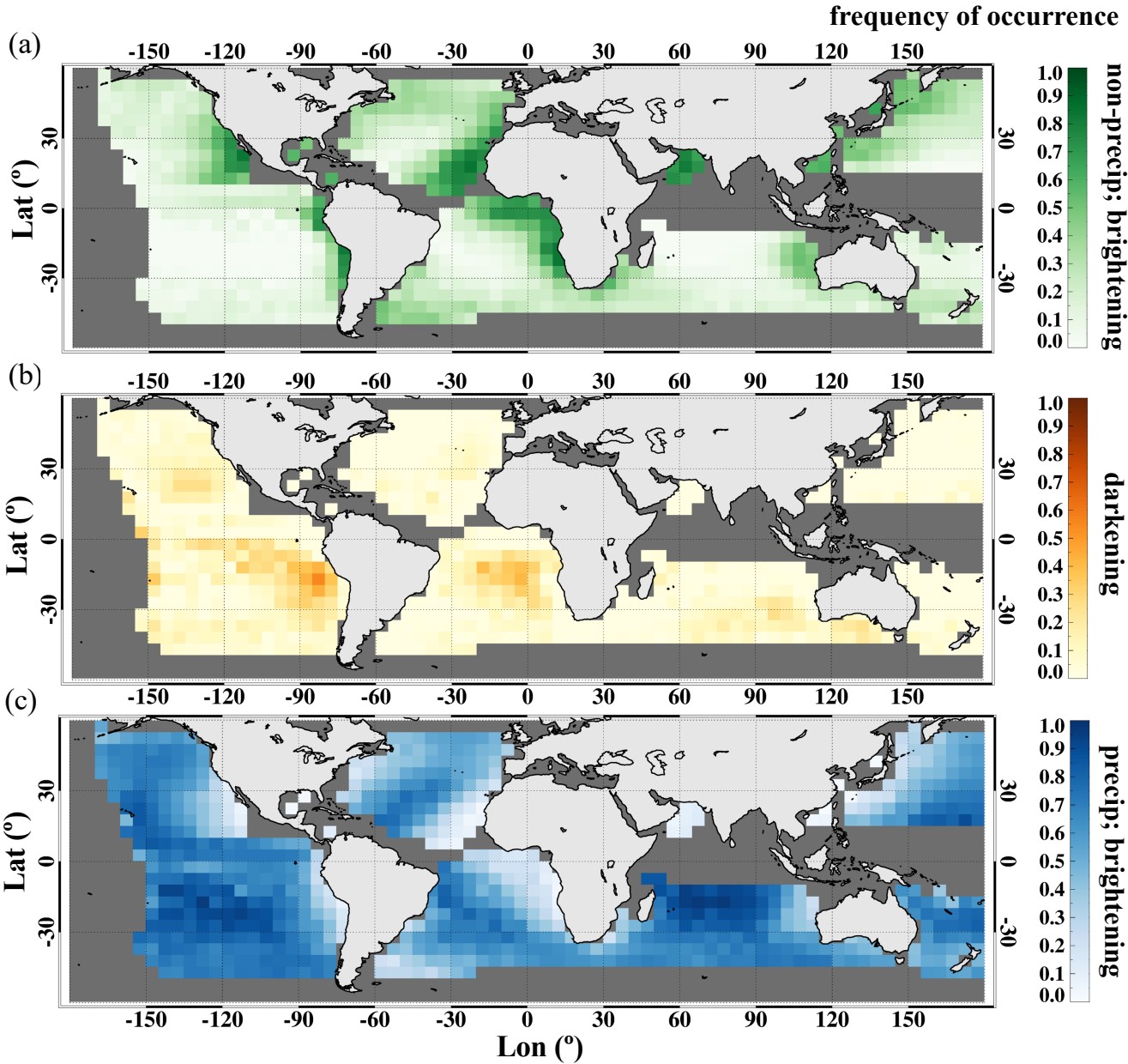

**Figure 3.** Geographical distribution of the frequency of occurrence of the 3 susceptibility regimes: (a) non-precipitating brightening, (b) darkening, and (c) precipitating brightening. The 3 regimes are separated based on the sign of $S_0$ and a $r_e$ of 12 $\mu$m in the LWP-$N_d$ variable space, for $5° \times 5°$ areas, similarly to Zhang et al. (2022). Only areas with SLLC frequency of occurrence greater than 0.1 are shown.

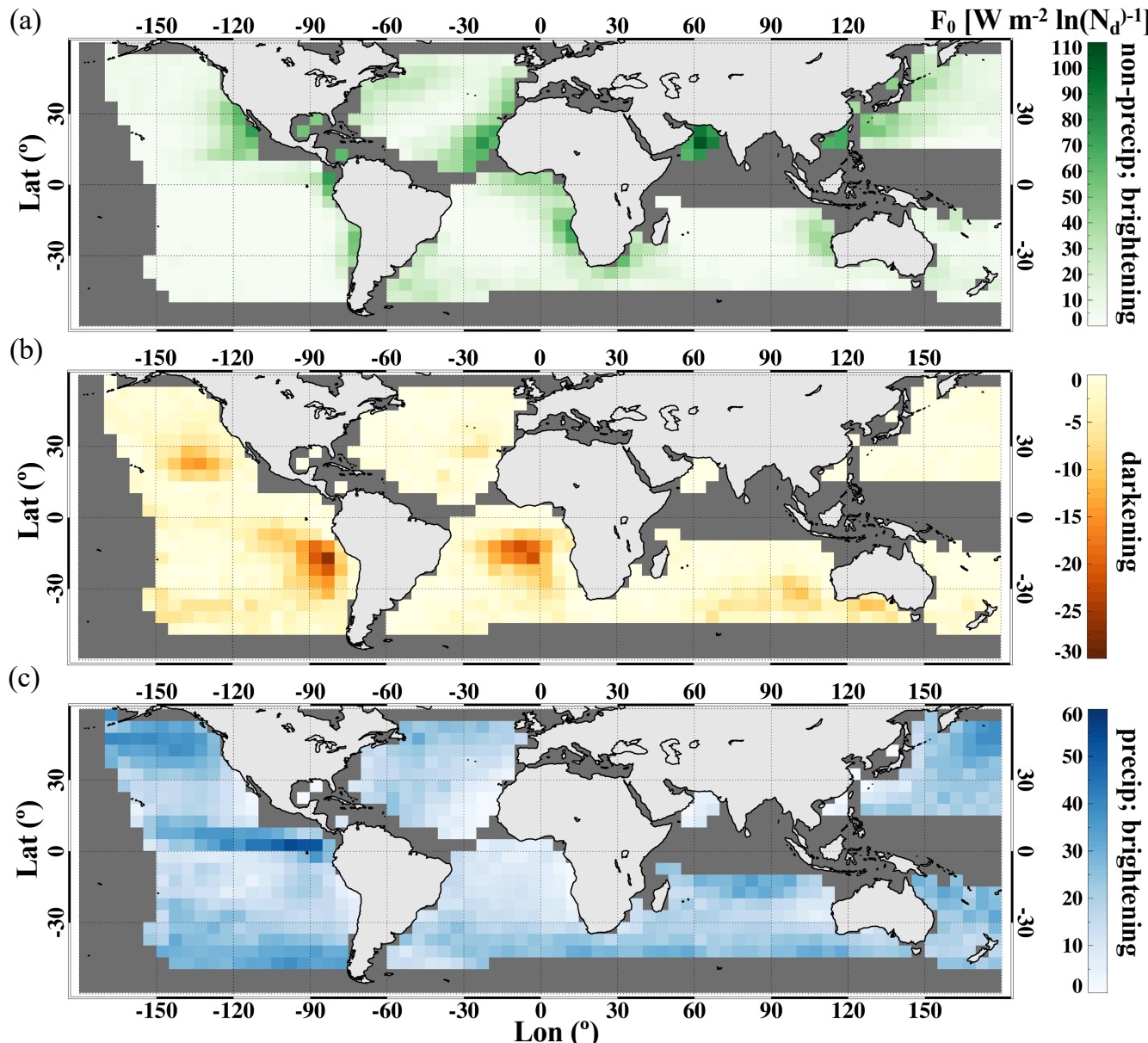

**Figure 4.** As in Fig. 3, but showing radiative susceptibility ($F_0$).

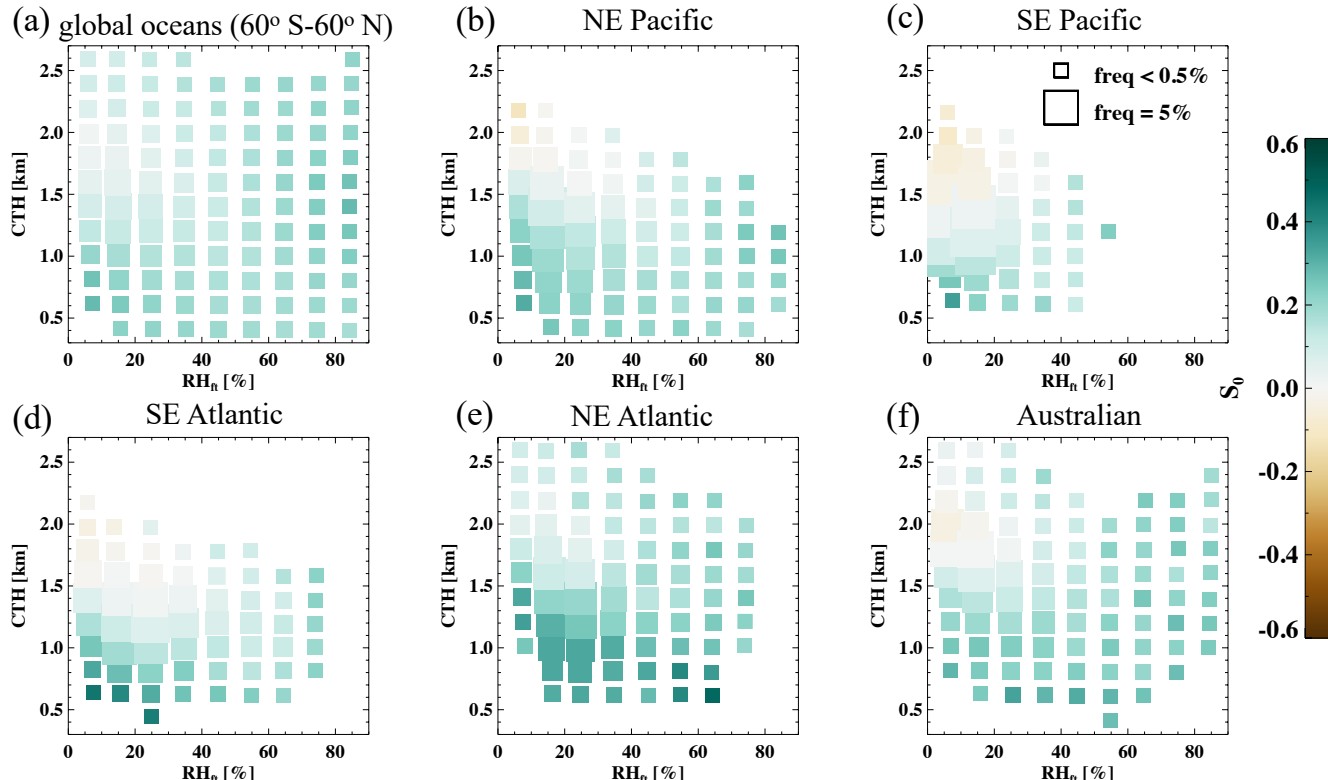

**Figure 5.** Mean $S_0$ under different meteorological conditions, namely free-tropospheric relative humidity (RH$_{ft}$; x-axis) and cloud top height (CTH; y-axis; a proxy for the marine boundary layer depth), for (a) global oceans (60° S - 60° N), (b) NE Pacific, (c) SE Pacific, (d) SE Atlantic, (e) NE Atlantic, and (f) Australian stratocumulus regions. Bin sizes for CTH and RH$_{ft}$ are 0.2 km and 10%, respectively. The size of the square indicates the frequency of occurrence of a meteorological state. Bins with less than 0.1% frequency of occurrence (or less than 100 samples) are not shown.

## 4    Distinct distributions of $S_0$ in 2-variable meteorological space at regional scale

Local adjustments of low clouds to aerosol perturbations are strongly dependent on the depth of the stratocumulus-topped
195  MBL (approximated by CTH; e.g. Possner et al., 2020; Toll et al., 2019), the strength of the capping inversion (indicated by LTS), and RH$_{ft}$ (e.g. Chen et al., 2014; Gryspeerdt et al., 2019). Figure 5-6 shows $S_0$ under different MBL and free-troposphere (FT) states, as a function of CTH, LTS, and RH$_{ft}$. Globally (60° S - 60° N), positive $S_0$ is found everywhere across the CTH–RH$_{ft}$ space, with less susceptible conditions occurring under drier FT and intermediate MBL depth (∼1.5 km; Fig. 5a). This is consistent with the SW heating and the entrainment feedback arguments (Section 3.2-2) that reduced droplet sizes lead to
200  stronger SW heating and entrainment mixing at cloud tops (Petters et al., 2012; Bretherton et al., 2007; Xue and Feingold, 2006), which is further facilitated by the deeper MBL and the drier air above cloud tops. As clouds become even deeper (>2 km), the likelihood of precipitation increases and the cloud brightening potential overwhelms the darkening potential (Fig.

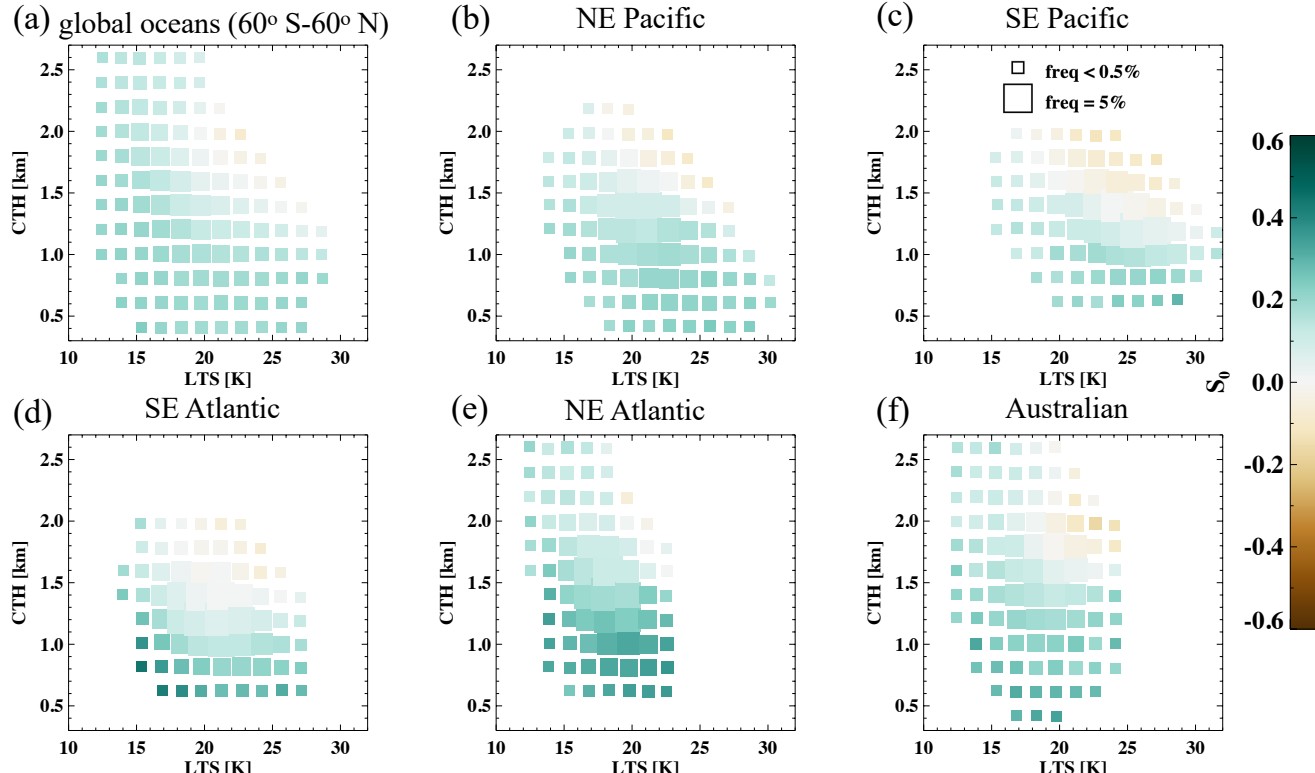

**Figure 6.** As in Fig. 5 but for CTH and Lower-Tropospheric Stability (LTS).

5). Similarly, weak brightening and darkening potential are associated with intermediate MBL depth and high LTS conditions over global oceans (Fig. 6), as expected, given that LTS anti-correlates with $RH_{ft}$ over subtropical oceans where large-scale free tropospheric subsidence prevails.

When the stratocumulus regime is singled out (Fig. 5b-f and Fig. 6b-f), the $S_0$ distribution in the two meteorological states is in qualitative agreement with the global analysis, however, cloud darkening (negative $S_0$) appears under the deep-MBL, dry-FT atmospheric states, more pronouncedly over the southeast Pacific stratocumulus deck (Fig. 5c), while weak brightening potential is observed under those conditions over the NE Atlantic (Fig. 5e). This is likely because of the lack of high-LTS conditions over the NE Atlantic (Fig. 6e). The prevalence of weak capping inversions allows the clouds to deepen and potentially precipitate, as opposed to entraining and drying with subsiding dry free-tropospheric air, evident over the other 4 basins (Fig. 6b-d, f). The "disappearance" of the darkening regime over the NE Atlantic, evident both in the LWP–$N_d$ space (Fig. 2) and in the meteorological spaces (Fig. 5-6), is consistent with the findings in Manshausen et al. (2022), where a slight increase in cloud LWP is found in "invisible" ship tracks, which are often identified under weaker cloud-top inversions compared to their surroundings.

Distinct "fingerprints" of $S_0$, characterized by the sign of $S_0$ (indicated by the colors) and the frequency of occurrence of meteorological states (indicated by the square sizes), in the large-scale meteorological factor spaces are evident, when individual basins are being compared (Fig. 5-6). This manifests in two ways; first, the frequency of occurrence of the FT and MBL conditions varies from basin to basin. For example, deep MBL (>2 km) or humid FT conditions rarely occur under the large-scale subsidence-dominated regions (Fig. 5b-d), compared to the NE Atlantic or the Australian basins (Fig. 5e-f) where high LTS conditions rarely occur (Fig. 6e-f). Second, different $S_0$, at least in magnitude, and in some cases even in sign, are observed across basins under similar conditions (e.g. CTH–RH$_{ft}$; Fig. 5). This suggests that cloud states are not necessarily the same under the same CTH–RH$_{ft}$ conditions, implying that other meteorological factors co-evolve with MBL and FT states differently from region to region, leaving distinct imprints on $S_0$. In the global analysis (Fig. 5a and 6a), these distinct regional "fingerprints" of $S_0$-meteorology relationships are not discernible due to the merging of different meteorology regimes.

## 5 Temporal covariabilities: Meteorology, albedo susceptibility, and aerosol conditions

### 5.1 Meteorological covariability

Four key large-scale meteorological factors evolve and co-vary distinctly across basins (Fig. 7, right column), leading to markedly different monthly evolution in $F_0$ (Fig. 7, left column). Even among regions strongly influenced by large-scale subsidence (Fig. 7a-c), large-scale meteorological conditions vary in magnitude and do not covary the same way temporally (e.g. RH$_{ft}$ tracks SST except over the SE Atlantic, LTS anti-correlates with SST except over the NE Pacific). As a result of the complex and distinct regional covariability in meteorological conditions, the temporal rise and fall of a single meteorological factor leads to markedly different responses in $F_0$ across basins. For instance, when LTS peaks over the Australian stratus region, $F_0$ is at its annual maximum (Fig. 7e, January). In contrast, over the SE Atlantic, the peak LTS season (September-October) corresponds to less susceptible conditions, whereas the most susceptible clouds of this region are found during a transition of large-scale conditions (i.e. SST decreases and LTS increases, June-July, Fig. 7c), during which the non-precipitating brightening regime occurs the most frequently (Fig. 8c). Taking CTH as another example, high CTHs (deep MBLs) lead to strong precipitating brightening over the NE Atlantic, whereas deep MBLs over the SE Pacific show very weak brightening potentials, due to relatively high stability and dry FT conditions, in striking contrast to the NE Atlantic (Fig. 7b and d). In other words, when $F_0$ or $S_0$ peaks, different combinations of large-scale meteorological conditions occur in different basins. Although certain environmental conditions are known to favor susceptible clouds, e.g. a humid free troposphere and/or strong LTS (Chen et al., 2014), one may not be able to find susceptible clouds under such conditions over some regions due to these regionally distinct meteorological covariabilities.

The covariability among large-scale meteorological factors over the SE Atlantic follows that over the SE Pacific, although the ocean surface is warmer, LTS is weaker, and the FT is moister in general over the SE Atlantic (Fig. 7c). This leads to qualitatively similar $F_0$ evolutions between the two basins; i.e., high $F_0$ during austral winter and low $F_0$ during austral summer. An exception occurs during late fall to winter (June-July), when precipitating clouds over the SE Pacific exhibit relatively weak positive $F_0$ whereas non-precipitating high $N_d$ clouds occur and exhibit strong $F_0$ over the SE Atlantic. This

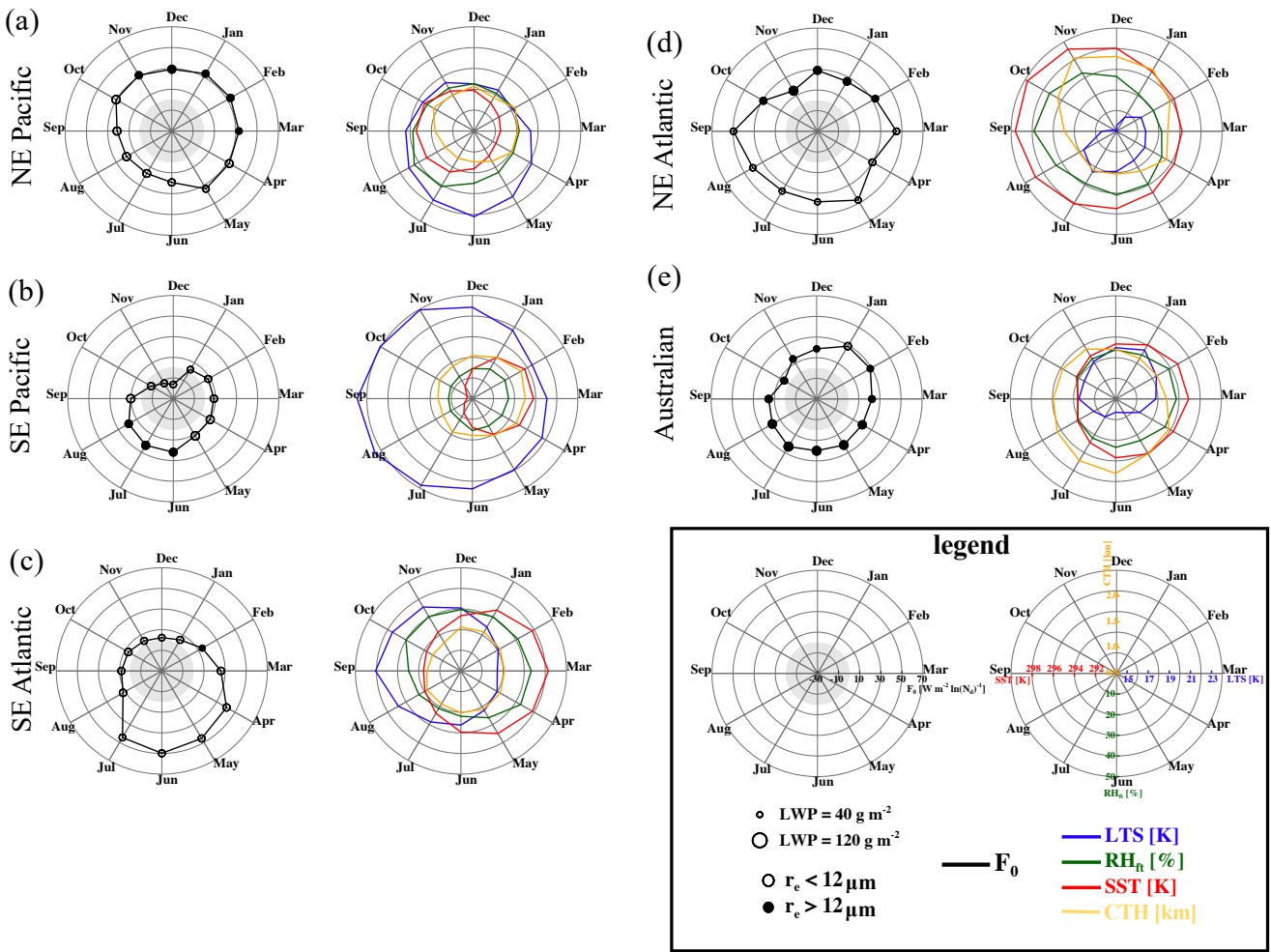

**Figure 7. Left column:** Monthly mean radiative susceptibility ($F_0$; black; positive values indicate cooling). Size of the circle indicates monthly mean LWP. Open (closed) circles indicate likely precipitating (non-precipitating) conditions, based on $r_e = 12~\mu m$. **Right column:** Monthly mean meteorological conditions: LTS (blue), $RH_{ft}$ (dark green), SST (red), and CTH (orange). Rows (a–e) represent results for the NE Pacific, SE Pacific, SE Atlantic, NE Atlantic, and the Australian stratocumulus regions, respectively.

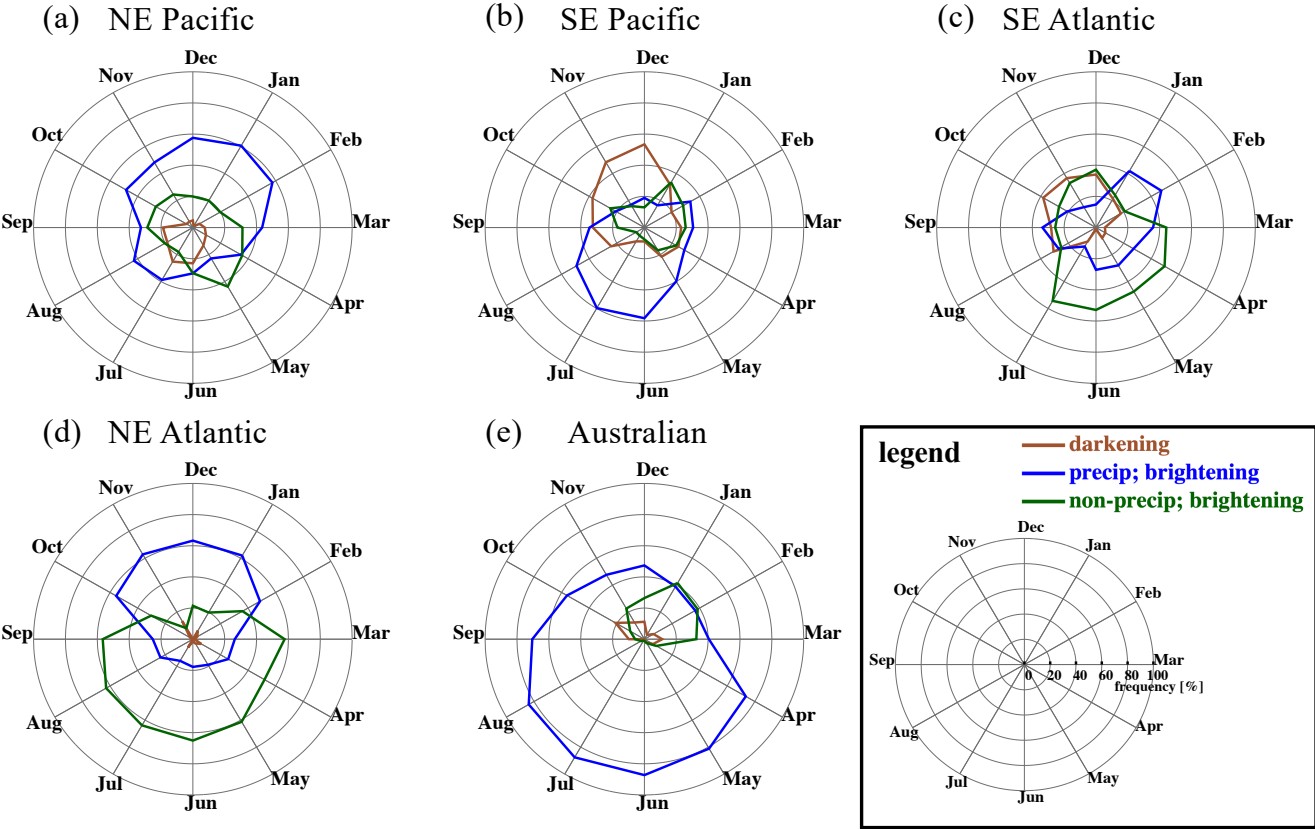

**Figure 8.** Monthly frequency of occurrence of the 3 susceptibility regime: precipitating brightening (blue), darkening (brown), and non-precipitating brightening (green). Rows (a–e) represent results for the NE Pacific, SE Pacific, SE Atlantic, NE Atlantic, and the Australian stratocumulus regions, respectively.

difference can be attributed to an aerosol source that is unique to the SE Atlantic basin, in the form of a large amount of biomass

burning aerosol that is advected by the co-occurring African Easterly Jet in the free troposphere during the southern African burning season (June-October; Adebiyi and Zuidema, 2016). The elevated aerosol is likely to be entrained into the MBL during June-July when the FT jet is not yet at its full strength (Zhang and Zuidema, 2021).

        Among five subtropical stratocumulus/stratus regions, the SE Pacific hosts the least susceptible conditions overall and is the only basin with monthly mean cloud darkening potential (Fig. 7b). This is consistent with the strongly subsiding (high

LTS) and extremely dry free troposphere observed therein, under which entrainment mixing at cloud tops can be extremely effective in reducing cloud LWP (thinning the clouds), and even more so when droplet sizes are reduced due to increasing aerosol. Low clouds over the NE Atlantic indicate the highest cloud brightening potential among the five regions, especially during March-September when the MBL is shallow and the free troposphere is relatively moist, giving rise to thin, non-precipitating clouds with low LWP and relatively high $N_d$ that exhibit brightening potentials (Fig. 7d). The lowest LTS (weakest

subsidence) among the 5 basins favors cloud deepening (an increase in LWP), leading to the lowest frequency of occurrence of the darkening regime over the NE Atlantic (Fig. 8d). During October-February over the NE Atlantic, when CTH is high (deep MBL), clouds precipitate often, leading to a frequently occurring precipitating brightening regime (Fig. 8d). Given the deep MBLs, precipitating conditions occur fairly frequently over the Australian stratus region almost throughout the year, except for the January-March period when increasing LTS leads to lower LWP and shallower MBL (Fig. 7e). As a result, the precipitating

brightening regime dominates almost all-year-round, with the non-precipitating brightening regime contributing only during austral summer (December-March) (Fig. 8e).

## 5.2 Albedo susceptibility and aerosol covariability

Albedo susceptibility ($S_0$), cloud frequency and areal coverage ($f_c$), incoming solar radiation ($SW_{dn}$), and aerosol perturbation ($\Delta \ln(N_d)$) collectively determine the SW flux change at TOA in response to an aerosol perturbation. Mathematically, if these

variables co-vary with each other in time, the temporal mean (indicated by the overbar) of their product

$$\Delta SW_{TOA} = \overline{S_0 \times f_c \times SW_{dn} \times \Delta \ln(N_d)} \tag{2}$$

will be biased if one takes temporal averages before multiplication.

$$\Delta SW_{TOA}^* = \overline{S_0} \times \overline{f_c} \times \overline{SW_{dn}} \times \overline{\Delta \ln(N_d)}. \tag{3}$$

Deriving $S_0$ from daily satellite snapshots enables us to resolve these variables temporally, giving us the opportunity to as-

sess the temporal covariability among them and how much it affects the mean (integrated over a long time period) SW flux changes at TOA. This has not been done in previous satellite studies that assess the $ERF_{aci}$ (most of them referenced in Bellouin et al. (2020)), mainly due to the use of the full temporal span of the dataset to derive a single value (with uncertainty range) for cloud or aerosol susceptibility. Here we assess the impact of temporal covariabilities on time-mean TOA SW flux responses from two perspectives: a) the $ERF_{aci}$ per unit anthropogenic emission, equivalent to $F_0$, and b) a marine cloud

brightening (MCB) experimental scenario where a prescribed $N_d$ perturbation of 300 cm$^{-3}$ is assumed, a value that might produce sufficient forcing to offset doubled $CO_2$ according to Wood (2021). We quantify percentage biases, $e$, calculated as $(\Delta SW_{TOA}^* - \Delta SW_{TOA})/\Delta SW_{TOA}$ at each 1° grid point, using daily values for each variable and considering a time-mean of 8-years (full data temporal range). Note the assessment in b) is not dependent on the value of the prescribed $\Delta N_d$ since percentage biases are being evaluated. Additionally, biases associated with using monthly mean values in $\overline{S_0 \times f_c \times SW_{dn} \times \Delta \ln(N_d)}$

are also evaluated, given that Global Climate Model (GCM) intercomparison projects store monthly mean outputs.

### 5.2.1 $ERF_{aci}$ per unit anthropogenic emission

In the first case, the $\Delta \ln(N_d)$ term is often estimated by the difference between present day and historical (pre-industrial) sulphate loading in the MBL (e.g. McCoy et al., 2017; Wall et al., 2022), which is not temporally resolved. Consequently, we only assess the bias associated with the covariability among $S_0$, $f_c$, and $SW_{dn}$, whose product is essentially $F_0$. Thus,

$F_0 = \overline{S_0 \times f_c \times SW_{dn}}$ and $F_0^* = \overline{S_0} \times \overline{f_c} \times \overline{SW_{dn}}$, where overbars indicate temporal averaging. Globally, $F_0^*$ overestimates $F_0$

| regions | $\langle e \rangle$ (%) | | $\frac{\langle e \rangle^2}{\langle e^2 \rangle}$ | |
|---|---|---|---|---|
| | monthly | 8-year | monthly | 8-year |
| global oceans ($60^0$S-$60^0$N) | 7.5 | 15 | $7.7\times10^{-5}$ | $1.1\times10^{-4}$ |
| NE Pacific | 4.9 | 9.2 | 0.51 | 0.63 |
| SE Pacific | -9.5 | 14 | 0.001 | $5.5\times10^{-4}$ |
| SE Atlantic | -1.4 | 3.0 | $4.1\times10^{-4}$ | $8.5\times10^{-4}$ |
| NE Atlantic | -0.7 | 1.0 | 0.002 | 0.003 |
| Australian | 6.0 | 8.1 | 0.42 | 0.36 |

**Table 1.** Percentage bias ($e$) in time-mean $F_0$ due to temporal covariabilities among $S_0$, $f_c$, and $SW_{dn}$. $e$ is calculated as $(F_0^* - F_0)/F_0$. $F_0^*$ is calculated using monthly mean and 8-year mean values. Mean of bias, $\langle e \rangle$, and the ratio between the squared mean bias, $\langle e \rangle^2$, and the mean squared bias, $\langle e^2 \rangle$, are shown for global oceans and the 5 stratocumulus regions. Angle brackets denote spatial averaging. Grid points with less than 30 samples and outside the 10[th] and 90[th] percentile of $e$, are excluded from the spatial averaging, to remove highly uncertain bias calculations.

by 15% (i.e. less cooling potential at TOA when covariability is considered by using temporally resolved variables), with the bias non-systematic geographically ($\langle e \rangle^2/\langle e^2 \rangle \sim 0$) (Table 1). For the five stratocumulus/stratus regions, the overestimation by $F_0^*$ is less, but varies greatly from basin to basin, 1% over the NE Atlantic to 14% over the SE Pacific, suggesting that the degree of covariability among $S_0$, $f_c$, and $SW_{dn}$ varies across basins. Worth noting is that the degree of geographically systematic bias also differs from basin to basin, such that 63% of the grid points over the NE Pacific show consistent sign in the bias, whereas non-systematic biases are found over the SE Pacific and SE Atlantic. This indicates that $S_0$, $f_c$, and $SW_{dn}$ temporally co-vary more systematically over the NE Pacific than the other basins. When monthly mean values are used for individual terms in $\overline{S_0 \times f_c \times SW_{dn}}$, biases decrease globally and at each basin, compared to that when 8-year means are used (Table 1), which suggests that monthly means are able to capture some degree of the temporal covariability among these variables. Interestingly, in 3 of the 5 basins (SE Pacific, SE and NE Atlantic), the sign of monthly mean biases flips, compared to that of 8-year means. This likely indicates that the temporal covariabilities among $S_0$, $f_c$, and $SW_{dn}$ are more pronounced at the monthly time scale. There is no noticeable difference in the degree of geographical systematic bias when monthly means values are used, compared to that when 8-year means are used. Overall, comparing to the other uncertainties associated with estimating $ERF_{aci}$, e.g. uncertainties in satellite retrievals and the historical $N_d$ level (Bellouin et al., 2020), biases associated with using monthly means or climatological means do not pose a pressing concern (Table 1).

### 5.2.2 MCB experimental scenario

However, this is not the case when an MCB experimental scenario is considered; here the temporal covariability between radiative susceptibility ($F_0$) and aerosol conditions (indicated by $N_d$; Fig. 9) substantially increases the bias in $\Delta SW_{TOA}$ if climatological values of each variable in Eqn. 2 are used (Table 2). This covariability is evident in annual cycles (monthly

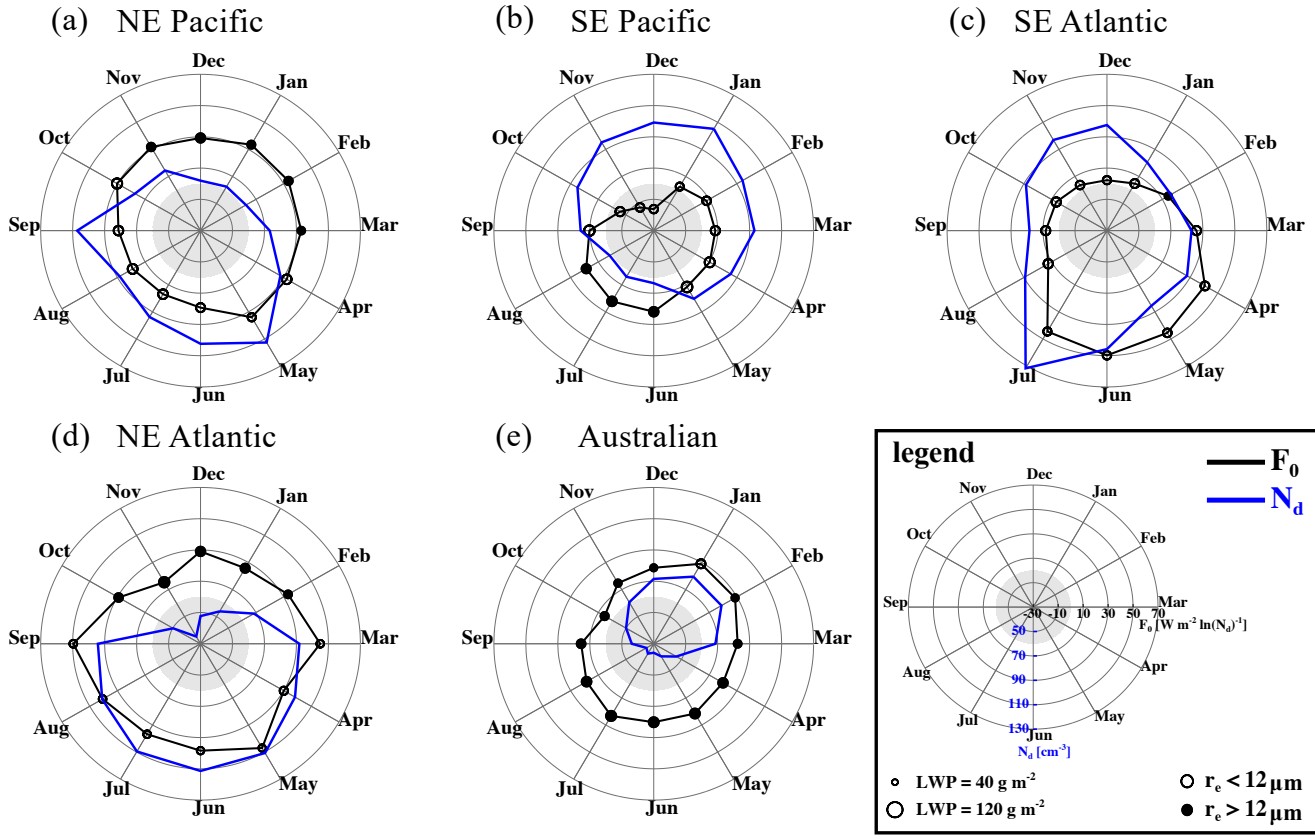

**Figure 9.** Same as in Fig. 7 left column, overlaid with monthly mean $N_d$ in blue.

| regions | $\langle e \rangle$ (%) | | $\frac{\langle e \rangle^2}{\langle e^2 \rangle}$ | |
|---|---|---|---|---|
| | monthly | 8-year | monthly | 8-year |
| global oceans ($60^0$S-$60^0$N) | -30 | -35 | 0.03 | 0.01 |
| NE Pacific | -41 | -46 | 0.97 | 0.97 |
| SE Pacific | -53 | -59 | 0.74 | 0.71 |
| SE Atlantic | -45 | -41 | 0.73 | 0.64 |
| NE Atlantic | -11 | -18 | 0.30 | 0.44 |
| Australian | -21 | -27 | 0.70 | 0.75 |

**Table 2.** As in Table 1, but with the percentage bias, $e$, calculated as $(\Delta SW^*_{TOA} - \Delta SW_{TOA})/\Delta SW_{TOA}$. $\Delta \ln(N_d) = \Delta N_d / N_d$, where $N_d$ is temporally resolved and $\Delta N_d$ is prescribed to be 300 cm$^{-3}$. Note the value of prescribed $\Delta N_d$ does not affect the results as percentage biases are evaluated.

evolutions) of $F_0$ and $N_d$ of each stratocumulus basin (Fig. 9). For instance, highly susceptible conditions (indicated by high $F_0$) co-occur with high background $N_d$, e.g. over the NE Pacific during May, over the SE Atlantic during July, over the NE Atlantic during May-July, and over the Australian basin during January (Fig. 9). This could lead to a muted $\Delta SW_{TOA}$, as the higher the background $N_d$, the lower the $\Delta \ln(N_d)$, when a constant $\Delta N_d$ is sustained by injecting aerosol particles into the MBL at a fixed rate, if such a MCB experiment were attempted. That said, $\Delta SW_{TOA}^*$ actually underestimates $\Delta SW_{TOA}$ by 35% globally (Table 2), meaning a stronger TOA SW flux perturbation is expected if the temporal covariability between albedo susceptibility and aerosol is considered. The reason is that although examples of high-susceptibility high-background-$N_d$ time period are evident, the overall covariability is dominated by the fact that low background $N_d$ conditions (high $\Delta \ln(N_d)$) are usually found under precipitating conditions where clouds exhibit precipitating-brightening potentials (Fig. 9). However, whether sustaining a constant $\Delta N_d$ (e.g. 300 cm$^{-3}$) is practical, especially under precipitation prevalent conditions, is outside the scope of this study, and one might need to resolve the diurnal cycle of these variables in order to study that.

Regionally, $\Delta SW_{TOA}^*$ underestimates $\Delta SW_{TOA}$ as well, ranging from -18% over the NE Atlantic to -59% over the SE Pacific. Clearly, biases are much higher in magnitude when the background $N_d$ dependent $\Delta \ln(N_d)$ term is included (Table 2). As mentioned above, percentage biases calculated here are not affected by the prescribed $\Delta N_d$ value. When monthly mean values are used, again, the magnitude of bias decreases globally and at each basin (except over the SE Atlantic), compared to that when 8-year means are used (Table 2). Nevertheless, substantial biases (more than 40%) associated with using monthly means are found over the 3 major stratocumulus decks. Worth noting is that biases are much more geographically systematic, indicated by the higher $\langle e \rangle^2 / \langle e^2 \rangle$ values, regardless of the degree of averaging (i.e. 8-year or monthly), compared to when $\Delta \ln(N_d)$ is not temporally resolved (Table 1). This suggests that temporal covariabilities are more geographically coherent at each basin.

These results stress, first, the necessity of taking the temporal covariability between albedo susceptibility and aerosol condition into account when the background $N_d$ becomes an influencing factor, e.g. for an MCB experimental scenario. Second, the temporal covariability among variables that determine $\Delta SW_{TOA}$ is basin specific, as biases vary in sign and magnitude across basins (Table 1 and 2). Regardless of the perspectives (ERF$_{aci}$ or MCB), the most pronounced bias is found over the SE Pacific, suggesting a strong temporal covariability among $S_0$, $f_c$, $SW_{dn}$, and background $N_d$ therein, whereas the NE Atlantic has the least such covariability.

## 6 Discussion

### 6.1 Diurnal cycle

$S_0$ and $F_0$, including the 3 susceptibility regimes, derived from the morning observations (Terra; Fig. S5-S8) have very similar geographical distributions as those observed in the afternoon (Aqua; Fig. 1-4), except a general shift towards positive $S_0$, as the darkening regime weakens and the precipitating brightening regime enhances with clouds occurring more frequently at higher LWP states (Fig. S6). One may find more noticeable differences in the cloud frequency weighted $F_0$ geographical distribution (Fig. S5b), especially over the remote part of the SE Pacific stratocumulus deck. This points to another layer of complexity in

characterizing and quantifying albedo susceptibility – that is the role of low cloud diurnal cycle, such that higher occurrence of low clouds in the morning over the SE Pacific (Garreaud and Muñoz, 2004) enhances the overall brightening potential therein. As a result, Terra observations indicate a slightly higher global mean occurrence-weighted $F_0$ of 3.1 W m$^{-2}$ ln($N_d$)$^{-1}$ (Fig. S6a), compared to 2.0 W m$^{-2}$ ln($N_d$)$^{-1}$ for Aqua observations (Fig. 2a). While the qualitative distributions of $S_0$ in the CTH–RH$_{ft}$ and the CTH–LTS state spaces remain the same, regardless of the observing time, the morning observations (Fig. S9-S10) indicate a general shift towards positive $S_0$ as well, which replaces the cloud darkening potentials related to the deep-MBL, dry-FT, and high-LTS conditions (Fig. 5-6) with a weak brightening potential. This, again, stresses the importance of cloud diurnal evolution for $S_0$ in that the same meteorological conditions may lead to opposing susceptibility regimes (i.e., brightening versus darkening) depending on the time of the day. Except for generally higher cloud LWP and higher $F_0$, the characterized meteorological covariabilities and the covariability among $N_d$ and $F_0$ for each stratocumulus basin using the Terra observations (Fig. S11-S13) agree well with those using the Aqua observations (Fig. 7-9). Although the seasonal trend in $F_0$ is the same between Aqua and Terra observations, the role of the diurnal cycle is manifested in the timing when monthly $F_0$ peaks (e.g. a lag of a month over the NE Pacific and the SE Atlantic; Fig. S13 and Fig. 9).

## 6.2 Implications

Our work is highly relevant to assessment of the radiative effect of aerosol-cloud interactions for climate applications. The approach we took enables us to reveal distinct $S_0$-meteorology-aerosol temporal covariabilities over 5 major subtropical marine stratocumulus/stratus decks, while yielding an integrated $F_0$ that is within the range of multi-evidence assessment of ERF$_{aci}$/$\Delta \ln(N_d)$ in Bellouin et al. (2020) (see Section 3.3).

In addition, our findings have direct implications for marine cloud brightening, which has been proposed as a way to mitigate the worst effects of the ongoing global warming crisis by creating more reflective (in the SW) MBL clouds, ideally with expanded areal coverage and prolonged lifetime, through deliberate aerosol injections (Latham et al., 2012). The bright, linear cloud features seen in satellite images, referred to as ship tracks (Coakley et al., 1987), are examples of ideal outcomes of an MCB experiment, and the conditionality of such an outcome on meteorological conditions is one of the key issues underpinning the viability of MCB. This study underscores two key points for the MCB community: 1) understanding or evaluating the impact of meteorology on cloud albedo susceptibility needs to be done at local/regional scales, where meteorological covariability is accounted for; 2) When scaling up the flux perturbation, it is crucial to consider the natural covariability between meteorology and aerosol, to which TOA SW flux responses to aerosol perturbation are sensitive. The latter point stresses the importance of shifting our attention from finding the most susceptible clouds to finding susceptible clouds that co-occur with favorable conditions (e.g. low background $N_d$, Fig. 9), as the amount of cooling at TOA due to cloud brightening depends strongly on both the albedo susceptibility and the background $N_d$ (e.g. Wang et al., 2011; Hu et al., 2021).

Although it is quite straightforward nowadays for one to assess how much of an aerosol perturbation is needed to achieve a certain increase in reflective SW using cloud-resolving simulation experiments (e.g. Wang et al., 2011; Chun et al., 2022), these modeling experiments are often state and scale specific (i.e. sensitive to the initial conditions of the simulations) and do not consider the frequency of occurrence of environmental conditions, nor the background aerosol concentrations, both of which

are crucial for scaling up the responses. Therefore, long-term satellite estimates of albedo susceptibility and flux perturbation that cover the full spatiotemporal frequency of occurrence of environmental conditions world-wide are an important asset for MCB research.

Another key implication of this study is that on the path towards understanding the influence of meteorology on aerosol-cloud interactions, efforts have been made to analyze the entire low cloud population collectively (globally) (e.g. Chen et al., 2014), while our study suggests such global analyses (e.g. Fig. 2a and Fig. 5a) do not always represent what happens regionally. In other words, while certain combinations of meteorological conditions appear to favor susceptible conditions in a global analysis, one may not be able to find such combinations at a given geographical location of interest, owing to the regionally 385  distinct meteorological covariability, and it is also likely that a different combination of meteorological conditions hosts the most susceptible clouds of that location.

## 7   Concluding Remarks

Marine warm cloud albedo susceptibility is derived from satellite-retrieved cloud microphysical properties and radiative fluxes, and sorted by day and geographical location. Geographical distributions of albedo susceptibility and the contributions from 390  three susceptibility regimes (non-precipitating brightening, darkening, precipitating brightening) are shown over global oceans (60° S to 60° N). Monthly evolutions in cloud radiative susceptibility, meteorological conditions (from ERA5 reanalysis), and cloud states (LWP and $N_\mathrm{d}$) are shown for five major eastern subtropical stratus/stratocumulus regions (20° × 20°), to illustrate the covariability among them and its impact on the response of TOA SW flux. The key findings are as follows:

1. An overall cloud brightening potential (positive $S_0$) is found in 8-year mean for most of global marine warm clouds –
395        most pronounced over subtropical coastal regions where shallow marine stratocumulus prevail along with relatively high annual-mean $N_\mathrm{d}$, and over the equatorial eastern Pacific where clouds rain more often (Fig. 1).

2. Cloud darkening associated with entrainment-driven negative LWP adjustments offsets the cloud brightening potential over remote parts of the stratocumulus regions where deeper MBLs favor cloud top entrainment, especially over the SE Pacific/Atlantic where darkening overcomes brightening in 8-year mean (Fig. 1 and Figs. 3-4).

3. The distinct regional "fingerprints" of $S_0$ in the LWP-$N_\mathrm{d}$ and CTH–RH$_{ft}$-LTS variable spaces are indiscernible in the global analysis because different low cloud and meteorology regimes are merged in a global analysis (Figs. 2 and 5-6).

4. Meteorological conditions have distinct regional covariabilities, leading to markedly different monthly evolutions in $F_0$ (Fig. 7).

5. The SE Pacific, a region with the driest free-tropospheric conditions and the highest LTS, hosts the least suscepti-
ble clouds exhibiting cloud darkening potential over several months during austral winter. Frequently occurring non-precipitating low-LWP, high-$N_\mathrm{d}$ clouds, found in shallow MBLs (March-September) over the NE Atlantic, represent the highest potential radiative responses to $N_\mathrm{d}$ perturbations among the five stratocumulus regions (Figs. 7-9).

6. While the qualitative agreement between Terra and Aqua underscores the robustness of our findings, their quantitative disagreement points to the important role of cloud diurnal evolution in determining albedo susceptibility (Figs. S5-S13).

7. Not only do various spatial-temporal averages applied in satellite-based approaches lead to biased $S_0$ (Feingold et al., 2022), non-negligible biases exist in $\Delta SW_{TOA}$ estimates, when the regionally distinct temporal covariability between $S_0$ and background $N_d$ is ignored by using a long-term mean values for each variable (Table 2).

8. In the search for the best targets for MCB, should such efforts be attempted, decisions should be made based not only on meteorological regimes, season, and time of day that produce the most susceptible clouds, but also the background $N_d$
(aerosol loading), which co-varies spatiotemporally with the susceptibility of the clouds (Fig. 9).

When the influence of meteorological conditions on low cloud $S_0$ are studied, it may seem tempting to try to disentangle effects of individual meteorological factors on $S_0$ by controlling for the others. Our results, however, indicate that this may not be the best approach since it is the natural covariability among meteorological conditions that dictates the regionally distinct temporal evolution in $S_0$. These results convey the importance of spatiotemporal variability in $S_0$ as a basis for both
understanding the limitation in scale-up of the meteorological influences on the radiative effect of aerosol-cloud interactions from regional to global, as well as for making decisions regarding when, where, and if marine cloud brightening efforts should be attempted.

*Data availability.* The CERES SSF data are publicly available from NASA's Langley Research Center (https://satcorps.larc.nasa.gov/) (Su et al., 2015). The fifth-generation ECMWF (ERA5) atmospheric reanalyses of the global climate data are available through the Copernicus
Climate Change Service (C3S, https://cds.climate.copernicus.eu/) (Hersbach et al., 2020).

*Author contributions.* JZ and GF designed the study. JZ carried out the analysis and wrote the manuscript. Both authors contributed to the interpretation of the results and finalizing the paper.

*Competing interests.* Graham Feingold is a co-editor of ACP. Other than this, the authors declare that they have no conflict of interests

*Acknowledgements.* We thank Johannes Mülmenstädt and another anonymous reviewer for their constructive comments and suggestions that
helped us improve the original paper.

*Financial support.* This research has been supported in part by the U.S. Department of Energy, Office of Science, Atmospheric System Research Program Interagency Award 89243020SSC000055, the U.S. Department of Commerce, Earth's Radiation Budget grant, NOAA CPO Climate & CI #03-01-07-001, and the NOAA Cooperative Agreement with CIRES, NA17OAR4320101.

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
