# Peer review of "Distinct regional meteorological influences on low cloud albedo susceptibility over global marine stratocumulus regions"

_EGUsphere, 2022_

## Referee Comment (RC1)

I have reviewed "Distinct regional meteorological influences on low cloud albedo susceptibility over global marine stratocumulus regions" by Zhang and Feingold.

In my opinion, the key advance in this manuscript is a global regime-based understanding of cloud albedo susceptibility to aerosols. We have known for a long time that aerosol–cloud interactions can be very different (even have different sign) depending on regime, but here the authors are drawing on a regime classification that divides clouds up by ACI mechanism: the Twomey-dominated "non-precip brightening" and the rapid adjustment-dominated "precip brightening" and "non-precip darkening". These regimes were developed in previous work by the authors, both process-understanding work in LES and observational work over one Sc region. What is new is a global perspective that shows that (a) different regimes dominate the ERFaci in different regions (near-shore Sc dominated by Twomey brightening, Sc to ShCu transition dominated by LWP adjustment dimming, and remote oceans dominated by LWP adjustment brightening) and (b) the covariability of meteorological controls in different regions, rather than individual meteorological controls on their own, modulate cloud susceptibility.

These results tie global albedo susceptibility (and thus ERFaci) to fundamental processes in a much more direct way than I have ever seen, bridging the gap between process studies (which are typically highly specific to a particular set of boundary conditions) and global observational studies (which often struggle with establishing causal relationships between aerosol and cloud properties because it is difficult to infer processes from observations). Many communities (including the global ERFaci, geoengineering, and process understanding communities) will take a keen interest. I recommend publication.

My one minor revision suggestion stems from l. 265, "Our work is highly relevant to assessment of the radiative effect of aerosol-cloud interactions for climate applications." I wholeheartedly agree with this statement. I think it would help the reader place these results in context if they (the results, not the reader) were compared to the Bellouin et al. (2020) ERFaci range. More concretely, eqs. (14) and (24) in Bellouin [sticklers might note that (24) is not an equation] lead me to believe that we should be able to compare the occurrence-weighted $F_0 = 32.9$ W m$^{-2}$ $(\ln N_d)^{-1}$ from Fig. 2a in the manuscript to the sum of RFaci and rapid adjustments per $\Delta \ln N_d$ in Bellouin, i.e.,

$$\text{ERFaci}/\Delta \ln N_d = S_N c_N + \beta_{\ln \mathcal{L} - \ln N_d} S_{\mathcal{L},N} c_{\mathcal{L}} + \beta_{\ln \mathcal{C} - \ln N_d} S_{\mathcal{C},N} c_{\mathcal{C}}. \tag{1}$$

These parameters are listed in Table 4 of Bellouin. (Parenthetically, I have plugged the central values from that table into (1) and get approximately $-9$ W m$^{-2}$ $(\ln N_d)^{-1}$. The disagreement in sign is a result of sign convention choices, but the factor 3 disagreement in magnitude puzzles me. I must be missing something obvious; perhaps the Bellouin number is all-sky and the manuscript number is cloudy-sky?)

I am also attaching an annotated copy of the manuscript with a number of minor typo corrections, typesetting pedantry, and suggestions that the authors should feel free to adopt or ignore.

[revised manuscript text omitted]

---

## Referee Comment (RC2)

This paper utilizes MODIS cloud observations and environmental variables from ERA-5 to analyze the meteorological influences on low-cloud susceptibility. They find that non-precipitative brightening typically occurs near the coast in primarily stratocumulus regions, precipitative brightening is most frequent over the central oceans, and darkening occurs predominately in the stratocumulus regions of the southeast Pacific and Atlantic (west of the non-precipitative brightening region). Regarding any meteorological influences, they found that the co-variability between the different variables analyzed influences the monthly evolution of albedo susceptibility, and differs depending on region.

Overall, I think this is a well written paper with impactful results, however I do have a few questions (listed below) that I would like answered prior to publication.

General Comments/Questions:

Lines 119 – 120: Regarding the precipitation brightening regime shown in Figure 2 and discussed here, "to the left of the 12-15 microns" means all susceptibility values to the left of the 12-micron isoline (not between the 12- and 15-micron isolines)?

Lines 124 – 126: You discuss how heavy precipitation would reduce cloud water through drop scavenging but state that your focus on high-cloud-fraction scenes does not allow you to analyze this (If I understand correctly). How do you think you results would have changed if you could have analyzed scenes where scavenging had occurred. Maybe I completely missed this point, but was curious about it.

Lines 160 – 163 and Figure 3: It looks like precipitation brightening is most frequent over the entire ocean basins other than right near the coast (especially in regions of primarily stratocumulus). Is this what you mean in the sentence starting "The precipitating brightening regime, although occurring over 50% of the time"?

Minor comments:

Lines 30 – 35: "Simulations of marine boundary layer (MBL) clouds," this sentence is a bit clunky (i.e. a bunch of comma splices) which reduces its readability. If you could break it up, that would be appreciated.

Line 76: What does $f_c$ represent in the radiative susceptibility equation?

Line 124: "Heavy precipitations deplete" should be "Heavy precipitation depletes"

Figure 6: It took me three reads to see the contour labels (i.e. lines of constant $F_o$) in the left plot of each panel (a-e). This made my interpretation of some of the text (e.g. lines 218 – 219: discussing that the southeast Pacific has the only monthly mean darkening potential) difficult.

---

## Author Comment (AC1)

**Response to reviews of "Distinct regional meteorological influences on low cloud albedo susceptibility over global marine stratocumulus regions" by J. Zhang and G. Feingold.**

We would like to thank Johannes Mülmenstädt and another anonymous reviewer for their encouraging feedback and constructive comments and suggestions on our manuscript, which helped us improve the original manuscript.

Specific responses to each comment are contained below, with the reviewers' comments provided in **blue** and our responses in **black**. Changes to the manuscript made in response to the reviewer are provided in ***red italics***. We have also made unsolicited changes to the manuscript to further polish the writing.

**REVIEWER 1 (Johannes Mülmenstädt):**

I have reviewed "Distinct regional meteorological influences on low cloud albedo susceptibility over global marine stratocumulus regions" by Zhang and Feingold.

In my opinion, the key advance in this manuscript is a global regime-based understanding of cloud albedo susceptibility to aerosols. We have known for a long time that aerosol–cloud interactions can be very different (even have different sign) depending on regime, but here the authors are drawing on a regime classification that divides clouds up by ACI mechanism: the Twomey-dominated "non-precip brightening" and the rapid adjustment-dominated "precip brightening" and "non-precip darkening". These regimes were developed in previous work by the authors, both process-understanding work in LES and observational work over one Sc region. What is new is a global perspective that shows that (a) different regimes dominate the ERFaci in different regions (near-shore Sc dominated by Twomey brightening, Sc to ShCu transition dominated by LWP adjustment dimming, and remote oceans dominated by LWP adjustment brightening) and (b) the covariability of meteorological controls in different regions, rather than individual meteorological controls on their own, modulate cloud susceptibility.

These results tie global albedo susceptibility (and thus ERFaci) to fundamental processes in a much more direct way than I have ever seen, bridging the gap between process studies (which are typically highly specific to a particular set of boundary conditions) and global observational studies (which often struggle with establishing causal relationships between aerosol and cloud properties because it is difficult to infer processes from observations). Many communities (including the global ERFaci, geoengineering, and process understanding communities) will take a keen interest. I recommend publication.

> We really appreciate these encouraging comments and glad to see that the impacts and implications of our work is being acknowledged and appreciated.

My one minor revision suggestion stems from l. 265, "Our work is highly relevant to assessment of the radiative effect of aerosol-cloud interactions for climate applications." I wholeheartedly agree with this statement. I think it would help the reader place these results in context if they

(the results, not the reader) were compared to the Bellouin et al. (2020) ERFaci range. More concretely, eqs. (14) and (24) in Bellouin [sticklers might note that (24) is not an equation] lead me to believe that we should be able to compare the occurrence-weighted $F_0 = 32.9$ W m$^{-2}$ (lnNd)$^{-1}$ from Fig. 2a in the manuscript to the sum of RFaci and rapid adjustments per $\Delta \ln Nd$ in Bellouin, i.e.,

$$\text{ERFaci}/\Delta \ln N_d = S_N c_N + \beta_{\ln \mathcal{L} - \ln N_d} S_{\mathcal{L},N} c_{\mathcal{L}} + \beta_{\ln \mathcal{C} - \ln N_d} S_{\mathcal{C},N} c_{\mathcal{C}}. \tag{1}$$

These parameters are listed in Table 4 of Bellouin. (Parenthetically, I have plugged the central values from that table into (1) and get approximately $-9$ W m$^{-2}$ (lnNd)$^{-1}$. The disagreement in sign is a result of sign convention choices, but the factor 3 disagreement in magnitude puzzles me. I must be missing something obvious; perhaps the Bellouin number is all-sky and the manuscript number is cloudy-sky?)

This is an excellent point, and we thank the reviewer for raising it, as it provides an opportunity to compare with the radiative sensitivity assessments discussed in Bellouin et al. (2020).

First, we think the approach the reviewer took trying to compare Fig. 2a with Bellouin20's number is a great one, and thereby, we adopted it in the revised manuscript.

However, there are 2 factors that we need to consider first in order to make a fair comparison:

1) We haven't assessed the cloud fraction adjustments in this study. (LCF adjustments can be very impactful in terms of radiative responses, and we're planning to look at it in the next study). Therefore, when comparing to B20, we do not consider their rapid adjustment in cloud cover term, i.e., we drop the last term in the reviewer's Eqn. 1 (above).

2) The occurrence-weighted $F_0$ we showed in the original Fig. 2a is actually weighted only by the freq of occurrence of individual LWP-Nd bins (i.e., they sum-up to almost 100%). Thus, the frequency of occurrence of these cloudy scenes has not yet been taken into account in these numbers, which is why they differ significantly from B20's assessment.

In order to make a fair comparison, we derived an effective cloud frequency factor, taking into account the spatial covariability between $F_0$ and the frequency of occurrence of high-fc scenes (scene$_{\text{freq}}$), which takes the form:

$$\text{freq}_{\text{eff}} = \frac{\langle \text{scene}_{\text{freq}} \times F_0 \rangle}{\langle F_0 \rangle}$$

Then, we scale the original occurrence-weighted $F_0$ values by this factor, globally and also regionally (separately derived). This puts us within the bounds provided in B20, and in fair agreement with B20 (cooling ~3.8 W/m2/ln(Nd), using their central values), with our Aqua assessment being ~ 2.0 W/m2/ln(Nd) and Terra assessment being 3.1 W/m2/ln(Nd). The difference likely stems from the focus of our study on marine-only high-fc scenes, whereas B20 integrates evidence from both modeling and satellite studies that cover global warm clouds.

We added a new sub-section (3.3 Scale-up of the $F_0$ assessment) to the revised manuscript to include this scale-up of our results to compare with Bellouin20.

I am also attaching an annotated copy of the manuscript with a number of minor typo corrections, typesetting pedantry, and suggestions that the authors should feel free to adopt or ignore.

These questions and editorial comments are super helpful and enlightening, as they have encouraged us think deeply about the implications of our results. We have incorporated them into the revised manuscript and responded to them individually (please see the attached annotated copy of the original manuscript with point-to-point responses). Tracked changes between the original manuscript and the revised version can be found in the resubmission package.

**REVIEWER 2:**

This paper utilizes MODIS cloud observations and environmental variables from ERA-5 to analyze the meteorological influences on low-cloud susceptibility. They find that non- precipitative brightening typically occurs near the coast in primarily stratocumulus regions, precipitative brightening is most frequent over the central oceans, and darkening occurs predominately in the stratocumulus regions of the southeast Pacific and Atlantic (west of the non-precipitative brightening region). Regarding any meteorological influences, they found that the co-variability between the different variables analyzed influences the monthly evolution of albedo susceptibility, and differs depending on region.
Overall, I think this is a well written paper with impactful results, however I do have a few questions (listed below) that I would like answered prior to publication.

We thank the reviewer for these encouraging comments and helpful suggestions.

General Comments/Questions:

Lines 119 – 120: Regarding the precipitation brightening regime shown in Figure 2 and discussed here, "to the left of the 12-15 microns" means all susceptibility values to the left of the 12-micron isoline (not between the 12- and 15-micron isolines)?

We apologize for this confusing statement, which should have been *"…to the left of the 12 micron"*, meaning all susceptibility values to the left of 12-micron isoline. This is corrected in the revised manuscript.

Lines 124 – 126: You discuss how heavy precipitation would reduce cloud water through drop scavenging but state that your focus on high-cloud-fraction scenes does not allow you to analyze this (If I understand correctly). How do you think you results would have changed if you could

have analyzed scenes where scavenging had occurred. Maybe I completely missed this point, but was curious about it.

Thanks for raising this question. We realized our sentences here weren't that clear and might mislead the reader.

Our point here is that heavy precipitating scenes do not represent a true cloud susceptibility to aerosol, and therefore we want to avoid these scenes (i.e., do not want to include them in our analysis). This is mostly ensured by focusing on high-fc scenes only, as heavily precipitating scenes are likely low fc scenes.

Here is our reasoning for this:
1) There are likely 3 processes occurring in precipitating stratocumulus scenes, a) precip-suppression (+ve LWP-Nd slope), b) precipitation scavenging (i.e. depleting LWP and Nd at the same time, → +ve LWP-Nd slope), and c) collision-coalescence & accretion (i.e. more LWP, less Nd, → -ve LWP-Nd slope).
2) What we really want to quantify, as susceptibility, is the precip-suppression process, not the other two.
3) Focusing on high-fc scenes certainly helped us minimize the likelihood of precip-scavenging, as this usually occurs with heavy precipitation, i.e., low fc; and process c) is certainly not strong enough to change the overall S0 here to -ve (Fig. 2).

Discussion around this is now revised and reads as *"The focus on high-fc scenes removes scenes with heavy precipitation, which is desirable since precipitation represents a cloud/rain effect on aerosol rather than an aerosol effect on cloud albedo."*

Lines 160 – 163 and Figure 3: It looks like precipitation brightening is most frequent over the entire ocean basins other than right near the coast (especially in regions of primarily stratocumulus). Is this what you mean in the sentence starting "The precipitating brightening regime, although occurring over 50% of the time"?

This is exactly the point we want to make.

In order to make this more clear, we rephrased it a bit to now read *"The precipitating brightening regime, although occurring over 50% of the time over most parts of the remote, clean oceans and the equatorial eastern Pacific (Fig. 3c), contributes little to the overall F0 (Fig. 4c)…"*

Minor comments:

Lines 30 – 35: "Simulations of marine boundary layer (MBL) clouds," this sentence is a bit clunky (i.e. a bunch of comma splices) which reduces its readability. If you could break it up, that would be appreciated.

Thanks for pointing this out! Indeed, it's a long and hard to follow sentence, which is now broken into 3 sentences in the revised manuscript.

Line 76: What does fc represent in the radiative susceptibility equation?

Sorry for the confusion; this 'fc' is now consistent with the symbol used for cloud fraction which is introduced earlier at the beginning of Section 2, after a typesetting correction. Thanks for pointing this out.

Line 124: "Heavy precipitations deplete" should be "Heavy precipitation depletes"

Thanks, and corrected.

Figure 6: It took me three reads to see the contour labels (i.e. lines of constant Fo) in the left plot of each panel (a-e). This made my interpretation of some of the text (e.g. lines 218 – 219: discussing that the southeast Pacific has the only monthly mean darkening potential) difficult.

Yes, indeed, we realized that these "spider-web" plots can be quite overwhelming when labels are overlaid with actual data curves, therefore we have revised the way we show these plots. We now remove all labels on the plot and have added a legend box to each figure (Figs. 7-9) showing empty "spider-webs" with labels but without any data. We believe these figures are now much easier to read.

I have reviewed "Distinct regional meteorological influences on low cloud albedo susceptibility over global marine stratocumulus regions" by Zhang and Feingold.

In my opinion, the key advance in this manuscript is a global regime-based understanding of cloud albedo susceptibility to aerosols. We have known for a long time that aerosol–cloud interactions can be very different (even have different sign) depending on regime, but here the authors are drawing on a regime classification that divides clouds up by ACI mechanism: the Twomey-dominated "non-precip brightening" and the rapid adjustment-dominated "precip brightening" and "non-precip darkening". These regimes were developed in previous work by the authors, both process-understanding work in LES and observational work over one Sc region. What is new is a global perspective that shows that (a) different regimes dominate the ERFaci in different regions (near-shore Sc dominated by Twomey brightening, Sc to ShCu transition dominated by LWP adjustment dimming, and remote oceans dominated by LWP adjustment brightening) and (b) the covariability of meteorological controls in different regions, rather than individual meteorological controls on their own, modulate cloud susceptibility.

These results tie global albedo susceptibility (and thus ERFaci) to fundamental processes in a much more direct way than I have ever seen, bridging the gap between process studies (which are typically highly specific to a particular set of boundary conditions) and global observational studies (which often struggle with establishing causal relationships between aerosol and cloud properties because it is difficult to infer processes from observations). Many communities (including the global ERFaci, geoengineering, and process understanding communities) will take a keen interest. I recommend publication.

My one minor revision suggestion stems from l. 265, "Our work is highly relevant to assessment of the radiative effect of aerosol-cloud interactions for climate applications." I wholeheartedly agree with this statement. I think it would help the reader place these results in context if they (the results, not the reader) were compared to the Bellouin et al. (2020) ERFaci range. More concretely, eqs. (14) and (24) in Bellouin [sticklers might note that (24) is not an equation] lead me to believe that we should be able to compare the occurrence-weighted $F_0 = 32.9$ W m$^{-2}$ $(\ln N_d)^{-1}$ from Fig. 2a in the manuscript to the sum of RFaci and rapid adjustments per $\Delta \ln N_d$ in Bellouin, i.e.,

$$\text{ERFaci}/\Delta \ln N_d = S_N c_N + \beta_{\ln \mathcal{L} - \ln N_d} S_{\mathcal{L},N} c_{\mathcal{L}} + \beta_{\ln \mathcal{C} - \ln N_d} S_{\mathcal{C},N} c_{\mathcal{C}}. \tag{1}$$

These parameters are listed in Table 4 of Bellouin. (Parenthetically, I have plugged the central values from that table into (1) and get approximately $-9$ W m$^{-2}$ $(\ln N_d)^{-1}$. The disagreement in sign is a result of sign convention choices, but the factor 3 disagreement in magnitude puzzles me. I must be missing something obvious; perhaps the Bellouin number is all-sky and the manuscript number is cloudy-sky?)

I am also attaching an annotated copy of the manuscript with a number of minor typo corrections, typesetting pedantry, and suggestions that the authors should feel free to adopt or ignore.

[revised manuscript text omitted]